# Trends in actinide electronic structure revealed from asymmetric, isostructural transuranic metallocenes
Cambell S. Conour[1,2,4], Mikaela Mary F. Pyrch[1,4], Nicholas J. Katzer[1,2], Asmita Sen[3], Megan R. Keener[1,2], Joshua J. Woods[2], Jochen Autschbach [3] ✉ & Polly L. Arnold [1,2] ✉

The study of actinide electronic structure and bonding within rigorously controlled environments is fundamental to advancing nuclear applications. Here, we report a new set of isostructural actinide organometallics; $An(COT^{big})_2$, ($An$ = Th, U, Np, and Pu), where $COT^{big}$ is the bulky 1,4-bis(triphenylsilyl)-substituted cyclooctatetraenyl dianion $(1,4-(Ph_3Si)_2C_8H_6)^{2-}$. The actinide(IV) metallocene sandwiches have a clam-shell structure, offering a new molecular symmetry to explore *f*-orbital contributions in bonding. Combined experimental and computational studies reveal that $An(COT^{big})_2$ complexes strongly differ from the previously published coplanar $An(COT)_2$ sandwiches due to the bent geometry and electron-withdrawing nature of the substituents. While $COT^{big}$ displays comparatively weaker electron donation, the low-energy *f-f* transitions in $An(COT^{big})_2$ have increased molar absorptivity consistent with the removal of the parity selection rule and better energetic matching between ligand and actinide *5f* orbitals as the series is traversed. For $Pu(COT^{big})_2$, covalent mixing of donor *5f* metal orbitals and the ligand-π orbitals is especially strong.

Understanding the role of *5f* and *6d* electrons in the electronic structure and reaction chemistry of actinides (An) is essential for nuclear waste separations and reprocessing[1]. Actinide chemistry occurs in a multitude of different environments, including the reducing conditions associated with deep geological storage, and is further complicated by the wide range of oxidation states available to the early actinides[2]. This demands a better understanding of actinide-ligand bond covalency in well-defined oxidation states and in varied coordination environments. While early actinide chemistry was historically dominated by aqueous and aerobic conditions, more recent efforts to study actinides in non-aqueous environments are increasingly providing new insights into the nature of An-ligand bonding[3,4].

To probe the trends in electronic structure across the *5f* period, a set of homoleptic, isostructural complexes is ideal. Yet, due to variations in metal size and preferred oxidation state across the periodic table, it is remarkably difficult to conserve a ligand set across any given row—particularly so for the large and poorly-understood actinide elements. Towards this end, $An(COT)_2$ complexes ($COT = C_8H_6^{2-}$), otherwise known as "actinocenes," continue to be of interest both experimentally and computationally. The high hapticity of the dianionic COT ligands, coupled with their ideal symmetry for metal *f*- and *d*-orbital interactions (including σ-, π-, δ-, and ϕ-bonding interactions)[3,4], renders them amenable to a wide range of specialized spectroscopic analyses such as X-ray absorption spectroscopy[5]. In fact, the only two series of crystallographically characterized, tetravalent An organometallic complexes containing Th, U, Np, and Pu are that of $An(COT)_2$[6-8] and $An(hdcCOT)_2$[9] ($hdcCOT = C_8H_4(C_3H_6)_2$). Notably, the $An(hdcCOT)_2$ system has recently been extended to include the first tetravalent berkelium (Bk) organometallic[10].

Detailed computational and experimental studies have revealed that in the $An(COT)_2$ series, the *6d* electron contribution to bonding remains nearly unchanged from Th to Pu, and the changes in covalency are assigned primarily to *5f* electron involvement[11,12]. This is in contrast to the pseudo-tetrahedral $An(Cp)_4$ complexes (Cp = cyclopentadiene)[13-16]. While $Pu(Cp)_4$ has not been crystallographically characterized, the series of $An(Cp)_4$ complexes has been the subject of rigorous computational study, revealing that the degree of *5f* participation in An-Cp bonding increases across the series, while the *6d* participation decreases slightly (with the *d*-shell occupation decreasing from 0.44 for Pa to 0.32 electrons for Pu)[13]. The similarities and differences between $An(COT)_2$ and $An(Cp)_4$ highlight the need for studies in additional coordination environments to explore the influence of the ligand geometry and electronics on An-C covalency for the early actinides.

Despite the wide range of functionalized COT systems[17-19], the U – Pu actinocenes consistently form a rigidly coplanar metallocene structure[20,21].

[1]College of Chemistry, University of California Berkeley, Berkeley, CA, USA. [2]Chemical Science Division, Lawrence Berkeley National Laboratory, Berkeley, CA, USA. [3]Department of Chemistry, University at Buffalo, State University of New York, Buffalo, NY, USA. [4]These authors contributed equally: Cambell S. Conour, Mikaela Mary F. Pyrch. ✉e-mail: jochena@buffalo.edu; pla@berkeley.edu

A "bent" metallocene is desirable as this conserves the coordination number while the loss of inversion symmetry allows for the increased mixing of the previously *ungerade* f-orbitals and *gerade* d-orbitals, leading to the potential for altered electronics within the An-C interactions. Additionally, the lack of inversion symmetry should enhance the intensity of observed *f-f* transitions[18]. Despite these desirable properties, the only bent, homoleptic actinocene analog is U(COT$^{big}$)$_2$, which was published with minimal characterization (COT$^{big}$ = 1,4-(Ph$_3$Si)$_2$C$_8$H$_6$)[22]. Herein, we report the synthesis and characterization of the bent actinide and transuranic metallocenes of COT$^{big}$ and systematically explore the impacts of *f*- and *d*-orbital contributions on covalency. Salt metathesis reactions of actinide(IV) chloride salts with K$_2$COT$^{big}$ enable the isolation of An(COT$^{big}$)$_2$ (An = Th, U – Pu). The series is characterized by UV-Vis and NMR spectroscopies and single-crystal X-ray diffraction (SCXRD). Additional characterization of the thorium and uranium compounds includes photoluminescence and IR spectroscopies. Spectroscopic characterization is supported by density functional theory (DFT) calculations, which identify electronic trends and are discussed in relation to structural features.

## Results and discussion

The synthesis of An(COT$^{big}$)$_2$ **1An** (An = Th, U, Np, and Pu) is best achieved by salt-elimination between the actinide tetrachloride (AnCl$_4$(DME)$_n$, An = U, n = 0; An = Th, Np, Pu, n = 2)[23–25] and the potassiated 1,4-*bis*(triphenylsilyl)-cyclooctatetraenide salt, K$_2$COT$^{big}$, in THF (Scheme 1). Further synthetic information and characterization of K$_2$COT$^{big}$ can be found in the Supporting Information (Figs. S2–S4, S10). This reaction can be performed with excess equivalents of ligand salt due to the high solubility of K$_2$COT$^{big}$ in THF, which enables the isolation of **1Np** and **1Pu** on single milligram scales. Crystallization *via* vapor diffusion of hexanes into toluene solutions of **1An** results in moderate crystalline yields (32%–78%). Additionally, vapor diffusion of hexanes into a benzene solution of each **1An** affords single crystals of **1Th**, **1U**, and **1Pu**, the SCXRD analyses of which show lattice benzene molecules. Diffraction data for the two uranium solvates (toluene and benzene) collected at different temperatures (100, 140, 240 K) show, as anticipated, no major influence of the lattice solvent on the structural metrics of **1An**, see SI for further analysis (Figs. S11–17 and Tables S1–3).

The $^1$H NMR spectra of toluene-$d_8$ solutions of **1An** show the presence of rotational isomers in solution for **1Th** and **1U** but only one isomer for **1Np** and **1Pu**. The $^1$H NMR spectra of **1Th** collected at 300 K are correspondingly broad for a diamagnetic species. At 350 K the resonances sharpen, and the resonances associated with the COT ring protons become more clearly defined (6.52, 6.99, 7.35 ppm, Figs. S5, S9), indicative of rotational isomerism. We assign the two isomers as *anti* and *gauche*; in the *anti*-conformer, the COT$^{big}$ rings are rotated 180° from being mutually eclipsed, and in *gauche*, the rings are rotated by approximately 98° from being mutually eclipsed (Fig. S1). Thus far, only the *gauche* configuration has been isolated in the solid state (see below), likely indicating the presence of stabilizing forces between SiPh$_3$ groups on opposite rings in the solids.

Similar to **1Th**, the $^1$H NMR spectrum of paramagnetic $f^2$ **1U** shows evidence of multiple isomers at 300 K; two sets of resonances for the three-ring C$_{Ar}$-H resonances in the range −7.9 to −62 ppm (−7.92, −19.23, −28.31, −49.04, −55.01, −62.27 ppm) and two sets of three broad SiPh$_3$ C$_{Ar}$-H resonances (0.56, 2.54 ppm (*ortho*); 4.54 ppm (*meta*), 5.33 ppm (*meta* & *para*, overlapping); Fig. S6). Between 300 and 350 K, coalescence of the *ortho* and *meta*-aryl SiPh$_3$ C$_{Ar}$-H resonances is observed for **1U** (Fig. 1). Comparison of solution phase UV-Vis (see below) and solid-state spectra of **1U** (from a crystalline sample determined to be exclusively in the *gauche* conformation, Fig. S23) display little to no difference in peak positions, indicating that the presence of the anti isomer in solution does not contribute extensively to the spectral analyses.

The $^1$H NMR spectrum of paramagnetic **1Np** displays broad, overlapping phenyl resonances, centered at 6.22 ppm, but resonances for the three-ring C$_{sp}^2$-H protons on the central COT ring are not resolvable (Fig. S7). The spectrum of **1Pu** is significantly sharper; both ring C$_{sp}^2$-H and

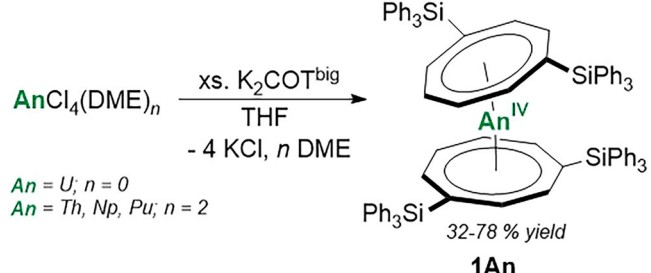

**Scheme 1** | Synthesis of **1An** (An = Th, U, Np, Pu).

SiPh$_3$ C$_{Ar}$-H resonances are resolvable and centered at 7.87 and 10.03 ppm, respectively (Fig. S8). These are assigned to the *gauche* isomer based on the solid-state structures as described previously.

Crystallographic studies show that **1An** are isostructural in the solid state. Crystals grow with the same rod morphology and are stable at room temperature in the absence of air and moisture. Structural data were collected at 240 K for **1Np** and **1Pu** due to radiological containment considerations, and thus **1Th** and **1U** diffraction data were also collected at the same temperature to enable meaningful comparison; selected metrics are in Table 1.

In the solid state, **1An** form a bent metallocene structure (Fig. 2). The four SiPh$_3$ groups interdigitate, and a space-filling model (Fig. S11) shows how effectively the ligands provide kinetic stabilization to the metal. In line with the actinide contraction, the An-COT$_{cent}$ distance decreases along the series from **1Th** to **1Pu**; 2.0128 (10) to 1.9198 (3) Å. The COT$_{cent}$-An-COT$_{cent}$′ angle is relatively constant across the series, ranging from 167.13(16)° (**1U**) to 168.74(9)° (**1Pu**). In comparison to An(hdcCOT)$_2$[9] and An(COT)$_2$[6–8], the An-COT$_{cent}$ distance in **1An** are consistently larger with **1Th** being larger by 0.007(1) Å (hdcCOT) and 0.009(1) Å (COT), **1U** by 0.018(2) Å (hdcCOT) and 0.024(2) Å (COT), **1Np** by 0.015(3) Å (hdcCOT) and 0.035(3) Å (COT), and **1Pu** by 0.008(2) Å (hdcCOT) and 0.035(2) Å (COT). Despite the bulkier substituents, the rate of contraction from Th to Pu across **1An** is similar to the other COT analogs.

The SiPh$_3$ groups are slightly bent away from the COT ring plane, but there is no discernible specific trend; the bend angles range from 8.90° (**1Th**) to 11.24° (**1Np**). The twist angle that defines how far the two C$_8$ rings are away from mutually eclipsed (*syn*) also appears uncorrelated with the An-COT$_{cent}$ distance, ranging from 95.30(14)° (**1U**) to 96.4(3)° (**1Np**). The COT rings are notably not perfect octagons, as is observed in the unsubstituted COT analogs, with C−C bonds varying from 1.394(11) Å (**1Np** C6−C7) to 1.448(10) Å (**1Np** C1−C2). The average C−C bond length does not decrease from **1Th** to **1Pu** (1.417(3) Å to 1.413(5) Å) within standard uncertainties. There are no substantial differences between the experimental structures of the complexes as extracted from the X-ray crystal structures and DFT-optimized molecular structures in the gas phase or solution (Fig. S27). Additionally, the experimentally observed trends in the An-COT$_{cent}$ distances and COT$_{cent}$-An-COT$_{cent}$′ angles are well-captured in the DFT-optimized geometries with the toluene solvent model (Table S4). This indicates that the structures of the complexes, as characterized crystallographically, are not strongly impacted by crystal packing.

The UV-Vis spectra of **1An** in toluene are shown in Fig. 3. The spectra modeled with time-dependent DFT (TPSSh functional) agree reasonably well with the experiments. For **1Th** and **1U**, the calculations show that the excitations around 3.5−3.8 eV have different assignments of their dominant components but reveal an associated similarity in the secondary components. For **1Th**, the intense peak near 3.75 eV (calcd. 3.8 eV) is assigned to a multi-component transition, which is different to the LMCT assignment made for the higher energy peaks (>3 eV) for Th(COT)$_2$ and Th(hdcCOT)$_2$[9]. This transition is more specifically assigned to a combination of COT-moiety π orbitals (*ungerade* parentage) combined with minor metal (Th) 5f contributions to the ligand phenyl substituents π* orbitals (Fig. S29). For **1U**, the electronic excitation calculated at 3.5 eV is

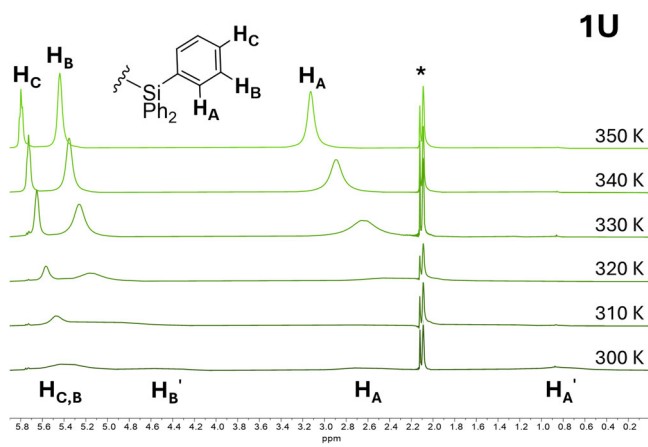

**Fig. 1 | Stacked variable temperature $^1$H NMR spectra of 1U in toluene-$d_8$ with resonances corresponding to the phenyl protons of the SiPh$_3$ groups labeled.** The temperature range is from 300 K (lower, dark green) to 350 K (upper, light green). The COT ring C$_{Ar}$-H ring resonances for **1U** are outside the displayed range, see SI for NMR spectra of all **1An**. * corresponds to the toluene CH$_3$ resonance.

**Table 1 | Selected bond metrics for 1An solid-state structures, measured at 240 K**

| Compound | M-COT$_{cent}$ distance (Å) | M-COT$_{cent}$ dist. (An radius subtracted) (Å) | Bend angle (°) |
|---|---|---|---|
| **1Th** | 2.0119 (7) | 0.9619 (7) | 167.61 (5) |
| **1U** | 1.948 (2) | 0.948 (2) | 167.13 (15) |
| **1Np** | 1.944 (3) | 0.964 (3) | 167.62 (19) |
| **1Pu** | 1.9198 (12) | 0.9598 (12) | 168.74 (9) |

The bend angles are calculated as the COT$_{cent}$-An-COT$_{cent}'$ angle, and the An radii correspond to the tetravalent, eight-coordinate Shannon radii[51].

assigned mostly as transitions from combinations of the COT-moiety π orbitals (*gerade* parentage) with admixture of U 6*d* to COT-moiety π* orbitals with minor U 5*f* admixture (Fig. S30). Closer inspection shows that the 3.8 eV **1Th** transition has a minor contribution (11%) with a similar assignment as the dominant component of the **1U** system (the *gerade* COT π/metal 6*d* combination to COT π*/metal 5*f*). Similarly, the transition in **1U** has a minor component (9%) that is similar to the major component of **1Th** (the *ungerade* COT π/metal 5*f* combination to COT phenyl π*).

As the series is traversed from Th to Pu, a gradual red-shift is observed in the corresponding bands, and they appear below 2.5 eV for **1Np** and **1Pu**. According to the analyses of the transitions (Figs. S30–33), for **1U**, **1Np**, and **1Pu**, the transition originates from the same type of orbital, i.e., combinations of COT-moiety π orbitals (*gerade* parentage) with admixture of An 6*d*. The acceptor orbitals of the transitions display COT-moiety π* character and have varying admixtures of metal 5*f*. The latter orbitals are stabilized energetically (Table S6) on moving from **1U** to **1Pu** because of the metal 5*f* shell energetic stabilization that is known to occur along the series; this correlates nicely with the red shifts observed for the relevant peaks in the calculated spectra and available experimental data.

The spectrum of $f^0$ **1Th** lacks characteristic features below 3 eV, i.e., at wavelengths longer than around 413 nm. Complexes **1U**, **1Np**, and **1Pu** possess 2, 3, and 4 electrons, respectively, in the 5*f* shell and correspondingly show low-intensity absorption features between 1 and 2.5 eV (Figs. S30, S32, S33). The lower-energy shoulder between 1.9 and 2.2 eV of the **1U** spectrum originates mainly from 5*f* to 6*d* transitions (Fig. 4), which would be electric dipole-allowed even in the presence of an inversion center. It is noted that the 5*f* – 6*d* transitions increase in energy from **1U** (1.5−2.5 eV) to **1Np** (2.8 eV) to **1Pu** (3.1 eV). This is attributed to the energy of the 6*d* orbitals being similar along the series while the energy of the 5*f* shell relative to 6*d* decreases with the increasing actinide nuclear charge, as noted already.

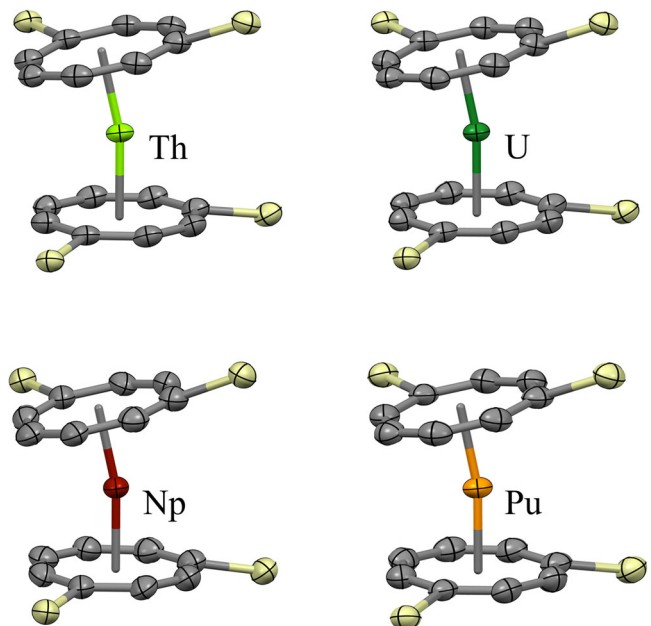

**Fig. 2 | Molecular structures of 1An with ellipsoids drawn at 50% probability.** Clockwise, starting with the top left: **1Th**, **1U**, **1Pu**, **1Np**. H atoms, Ph groups, and lattice solvent molecules are omitted for clarity. Atom key: light green, Th; green, U; maroon, Np; orange, Pu; gray, C; pale yellow, Si.

While the absorbance of An(hdcCOT)$_2$ complexes was not reported beyond ~700 nm, comparisons can be made between **1An** and An(COT)$_2$. **1U** and **1Np** display low-intensity (<100 cm$^{-1}$M$^{-1}$), low-energy (≥750 nm) absorbance peaks, corresponding to ligand sensitized *f-f* transitions (Fig. 4), which are not observed in the corresponding An(COT)$_2$ spectra. The intensity of these *f-f* transitions is likely enhanced by the perturbation of the inversion symmetry that exists in the original, coplanar COT metallocene series, eliminating the parity selection rule that forbids *f-f* transitions. Likewise, the low-energy *f-f* transition observed for **1Pu** has increased molar absorptivity compared to the other Pu complexes. This is potentially attributed to the heavy mixing of donor 5*f* metal orbitals and the ligand-π orbitals, giving the transition some LMCT character (Fig. S33), which is likely enhanced by a better matching of the energies of the actinide 5*f* orbitals and ligand orbitals as the actinide series is traversed. Notably, a similar absorbance is not observed for the Pu(COT)$_2$ and Pu(hdcCOT)$_2$ systems.

Because the assignment of the peaks in the **1Th** spectrum differs from that in Th(COT)$_2$ and Th(hdcCOT)$_2$ complexes, it is not straightforward to rationalize the minor blue shift of the 3.75 eV peak observed for **1Th** compared to the unsubstituted COT analog[9]. The red-shift of the bands on moving from Th(COT)$_2$ to Th(hdcCOT)$_2$ was rationalized by Russo and coworkers by the corresponding raising of the energies of the highest occupied ligand orbitals in the complexes[9]. Further direct comparisons between the assigned transitions of different COT derivatives were not made due to the very high level of computational demand that this would incur.

Overall, the introduction of 1,4-bis(triphenylsilyl) substitution to the COT ligand in the **1An** complexes is seen to have notable impacts on the resulting electronic structures. Firstly the occupied frontier canonical MOs of **1An** display qualitatively similar characteristics among themselves and with respect to the COT analogs, evidencing donation from the highest occupied ligand orbitals to the metal centers (Fig. 5 and Figs. S34, S35). In comparison to the unsubstituted COT, COT$^{big}$ displays weaker electron donation to the metal center, as is seen in the significantly higher calculated charge for **1An** compared to An(COT)$_2$ in Table 2. The same Table also shows that the degree of electron donation from COT and COT$^{big}$ to the metal overall is greatest for U/Np, then weaker to Pu, and weakest to Th. However, it is worth noting that the donation into the 5*f* shell, as reflected in the excess population of the 5*f* shell (relative to the formal count of 0, 2, 3, 4

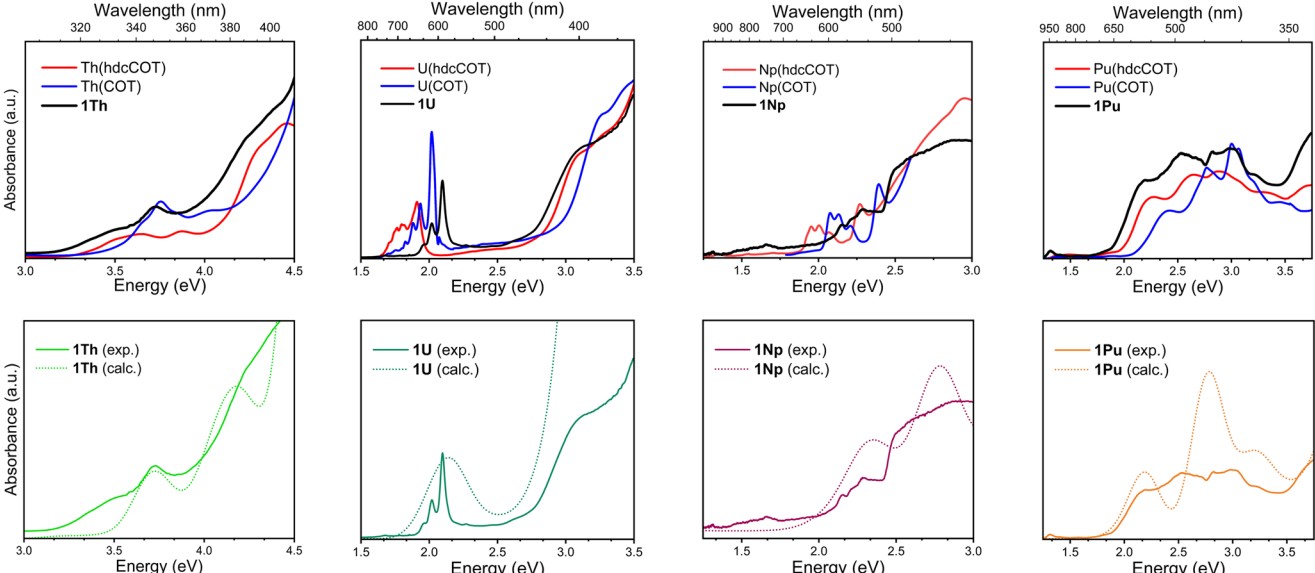

**Fig. 3 | Top: normalized UV-Vis spectra of 1Th, 1U, 1Np, and 1Pu in toluene compared to the analogs An(COT)$_2$ and An(hdcCOT)$_2$.** U-Th(COT)$_2$ and U-Th(hdcCOT)$_2$ spectra were previously reported in THF, while Np-Pu(hdcCOT)$_2$ spectra were collected in *n*-hexane[9]. The Np(COT)$_2$ absorbance spectrum in toluene was digitally extracted from previous reports[8]. The Pu(COT)$_2$ absorbance spectrum in toluene was reproduced with permission from the original authors[52]. Bottom: comparison of experimental (exp.) and calculated (calc.) absorption spectra within the region of interest. The computed spectra of **1Th**, **1U**, and **1Np** are blue-shifted by 0.15 eV, while **1Pu** is red-shifted by 0.3 eV. All spectra have been scaled for visual clarity and are displayed in arbitrary units (a.u.).

for Th, U, Np, Pu, respectively) increases along both series, whereas the donation to 6*d* is maximal at U, such that a non-monotonic trend in the overall calculated metal charge results. Uncovering the precise reasons for these trends would require a detailed analysis that is beyond the scope of this report, but they are likely associated with the underlying interplay of trends in the energies and overlaps of the participating metal and ligand atomic orbitals[26].

In unsubstituted COT, the highest energy occupied orbitals are 2-fold degenerate and responsible for much of the donation to the metal. The unsymmetric ligand substitutions and bent coordination environments in **1An** cause the highest donor levels to split in energy, and the ligand-to-metal donation is reduced relative to unsubstituted COT. The accompanying changes in the electronic spectra of **1An** can be attributed in part to the weaker donation and the electron accumulation in the ligand donor levels. The **1Th** system shows observable spectral contributions from transitions involving the COT$^{big}$ ligand aryl groups. Transitions into similar acceptor orbitals for the other **1An** appear in the calculations as well, but at comparatively high energies.

In addition to UV-Vis spectroscopy, we investigated the photoluminescence spectra of **1Th** and **1U**, as well as K$_2$COT$^{big}$. **1Th** exhibits a strong emission peak at 526 nm, with notable shoulders at 448 and 482 nm corresponding to a single excitation peak at 364 nm (Fig. S26). This is consistent with a predominantly ligand-centered emission[27,28], where the potassium salt exhibits emission features at 500, 540, and 584 nm (Fig. S25). The spacing of the peaks observed in K$_2$COT$^{big}$ and the shoulders in **1Th** (ca. 1425 cm$^{-1}$) is consistent with vibronic coupling to the vibrational mode observed at 1427 cm$^{-1}$ in the FTIR spectra of both compounds (Fig. S24). As is common for U(IV), **1U** displays no observable emission when excited at wavelengths between 250 nm and 600 nm, in line with the availability of non-radiative decay pathways involving the uranium 5*f*/6*d* manifold[29].

## Conclusions
We have synthesized a new suite of transuranic organometallic complexes and characterized them with structural and electronic studies assisted by computational (DFT) efforts. Due to the bulk of the COT$^{big}$ ligand, **1An** has substantially bent solid-state structures, deviating by up to 12.87° from the coplanarity displayed by the unsubstituted COT analogs. In solution, the $^1$H NMR spectra of **1Th** and **1U** show rotational isomerism, but only one

isomer for **1Np** and **1Pu** is observed, likely due to the contraction of the ligand around the smaller metal centers. These unsymmetric COT$^{big}$ complexes have notably perturbed electronic structures in comparison to their coplanar counterparts due to both ligand geometry and silyl substituents. In comparison to unsubstituted COT, COT$^{big}$ displays weaker electron donation to the metal center. Although the degree of electron donation from COT and COT$^{big}$ follows the same trend (**1U≈1Np > 1Pu > 1Th**), the amount of ligand-to-metal electron donation is more similar in **1U** and **1Pu** than in the corresponding COT analogs. Complementary work is in progress to develop further, lower-symmetry analogs of the canonical An(COT)$_2$ complexes, to further understand the roles of the different *f*- and *d*-orbitals in *f*-block metal-ligand bonding.

## Methods
### Caution
Uranium-238 and thorium-232 are low-level alpha emitters. All work with these isotopes was performed in monitored fume hoods or in inert gloveboxes in a radiological laboratory with portable α- and β-counting monitors. Neptunium-237 and plutonium-242 undergo alpha decay to protactinium-233 and uranium-238, respectively. The plutonium-242 stock solution used was 99.98 w/w% plutonium-242, accounting for 84% of the α emission. The remaining sources of α emission were $^{240}$Pu (0.02 w/w%, 0.76% α), $^{238}$Pu (<0.01 w/w%, 14.66% α), and $^{244}$Cm (<0.01 w/w%, 1.1% α). All transuranic (Z > 92) radioactive materials were handled in a radiological laboratory equipped with portable α- and β-counting monitors, high-efficiency particulate air (HEPA) filtered fume hoods, and negative-pressure gloveboxes equipped with HEPA filters. Appropriate personal protective equipment, body and extremity dosimetry, and a Canberra Sirius 5PAB hand-and-foot personal contamination monitoring station at the entrance/exit of the laboratory were used to ensure researcher safety. Daily and weekly surveys are performed to monitor for contamination of the laboratory. All manipulations involving dispersible transuranic materials as free-flowing solids were performed in a negative-pressure glovebox.

Unless otherwise noted, all manipulations were conducted under argon utilizing standard Schlenk techniques or under nitrogen in an MBraun UniLab Plus glovebox. All glassware, canulae, and Whatman 0.7 μm retention glass microfiber filters were dried for at least 16 h in a 175 °C oven and cooled under a dynamic vacuum prior to use. Molecular

**Fig. 4 |** Selected donor and acceptor NTOs (toluene solvent model) responsible for the lower-energy transitions in **1U** and **1Np**.

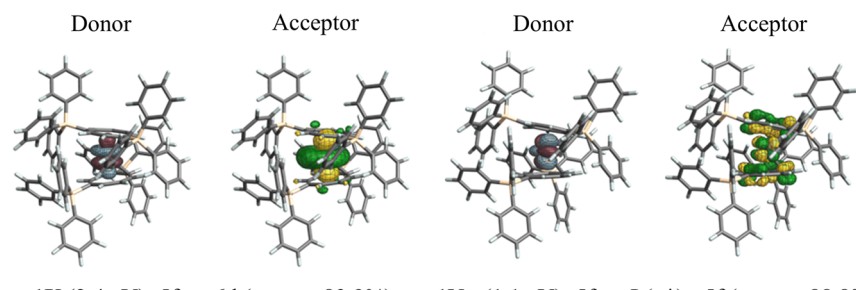

Donor　　　Acceptor　　　Donor　　　Acceptor

**1U** (2.4 eV): 5f → 6d (α → α, 83.9%)　　　**1Np** (1.1 eV): 5f → L(π*) +5f (α → α, 88.8%)

**Fig. 5 |** α-Spin HOMO and LUMO of **1An**.

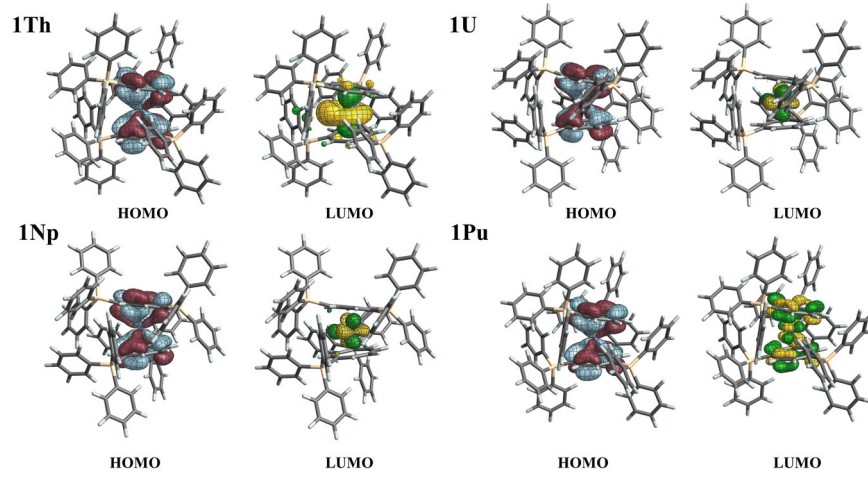

**Table 2 | PBE0 calculations with toluene solvent model**

| Compound | Natural charge on metal | Combined natural charges on the COT moiety carbons | Actinide effective electron configuration |
|---|---|---|---|
| **Th(COT)₂** | 1.00 | −5.30 | [Rn]$7s^{0.06}$ $5f^{1.05}$ $6d^{1.69}$ $7p^{0.03}$ $8s^{0.01}$ |
| **U(COT)₂** | 0.34 | −4.51 | [Rn]$7s^{0.06}$ $5f^{3.40}$ $6d^{1.74}$ $7p^{0.03}$ $8s^{0.01}$ |
| **Np(COT)₂** | 0.34 | −4.48 | [Rn]$7s^{0.06}$ $5f^{4.54}$ $6d^{1.66}$ $7p^{0.04}$ $8s^{0.01}$ |
| **Pu(COT)₂** | 0.57 | −4.67 | [Rn]$7s^{0.06}$ $5f^{6.55}$ $6d^{1.54}$ $7p^{0.06}$ $8s^{0.01}$ |
| **1Th** | 1.34 | −6.07 | [Rn]$7s^{0.08}$ $5f^{1.03}$ $6d^{1.33}$ $7p^{0.04}$ $8s^{0.02}$ |
| **1U** | 0.66 | −5.41 | [Rn]$7s^{0.07}$ $5f^{3.20}$ $6d^{1.51}$ $7p^{0.07}$ $8s^{0.02}$ |
| **1Np** | 0.61 | −5.33 | [Rn]$7s^{0.08}$ $5f^{4.47}$ $6d^{1.42}$ $7p^{0.05}$ $8s^{0.02}$ |
| **1Pu** | 0.75 | −5.45 | [Rn]$7s^{0.08}$ $5f^{6.52}$ $6d^{1.32}$ $7p^{0.10}$ $8s^{0.02}$ |

Natural charges and effective electronic configuration (reflecting donation) of the metal and total natural charges on the COT moiety in Th(COT)₂, and **1An** (An = Th, U-Pu) complexes from Natural Population Analysis (NPA)

sieves were preactivated by drying in a 175 °C oven for 48 h and further activated by microwave heating, then cooled under a dynamic vacuum. NMR spectra were recorded on either a 300 MHz Bruker AVANCE NEO equipped with a 5 mm BBO probe or a 500 MHz Bruker Avance IV NEO equipped with a 5 mm BBO Prodigy CryoProbe at 298 K unless stated otherwise. [1]H and [13]C spectra were referenced internally to the solvent residual and chemical shifts reported in ppm vs tetramethylsilane (0.0 ppm). [29]Si spectra were recorded using INEPT pulse sequences, were referenced internally to hexamethyldisiloxane, and chemical shifts were reported in ppm vs tetramethylsilane (0.0 ppm).

Benzene (bz), hexanes, tetrahydrofuran (THF), and toluene (tol) were purified using an MBraun Solvent Purification System with packed alumina columns, degassed, and stored over activated 4 Å molecular sieves for 24 h prior to use. Toluene-d₈ was refluxed over potassium, distilled, degassed, and stored over activated 4 Å molecular sieves for a minimum of 24 h prior to use. ThCl₄(DME)₂[23], UCl₄[24], NpCl₄(DME)₂[25], PuCl₄(DME)₂[25], and 1,4-

bis(triphenylsilyl)-cycloocta-2,5,7-triene (H₂COT[big])[22] were prepared according to literature procedures.

Complexes **1An** (An(1,4-(Ph₃Si)₂C₈H₆)₂(C₇H₈)) were synthesized following the same general procedure: K₂COT[big] (2.2−2.5 equiv.) was prepared as a homogenous THF solution (500 µL) and combined with a solution of AnCl₄(DME)₂ (An = Th, Np, Pu: n = 2; An = U: n = 0) in THF (500 µL). After stirring for *x* hours (**1Th**, **1U**: *x* = 4; **1Np**, *x* = 16, **1Pu**, *x* = 24), the mixture was centrifuged and the yellow supernatant containing excess ligand salt was separated by decantation to yield the target compound as a powder pellet. Washing to remove traces of residual ligand salt with THF (3 × 250 µL) afforded the product **1An** as a solid, which was dissolved in toluene, heated to 90 °C for approximately 30 min, centrifuged, and concentrated to yield pure target product as a powder. Single crystals suitable for X-ray diffraction were obtained by vapor diffusion of hexanes into a toluene solution of **1An**. For **1Th**, **1Np**, and **1Pu**, crystals of **1An·bz** were obtained by dissolving a small amount of **1An** (approximately 0.5 mg) into

1 mL of benzene, followed by vapor diffusion of hexanes into the mixture. Extended experimental details for all complexes can be found in the supporting information under the Supplementary Methods heading.

**1Th (Th$^{IV}$[1,4-(Ph$_3$Si)$_2$C$_8$H$_6$]$_2$(C$_7$H$_8$)):** 68% yield as a yellow powder (54 mg, 34 μmol), $^1$H NMR (300 MHz, d$_8$-toluene, 350 K) δ (ppm): 6.52 (2H, m, C$_{Ar}$-H), 6.99 (2H, d, C$_{Ar}$-H), 7.07 (18H, m, C$_{Ar}$-H), 7.33, (2H, d, C$_{Ar}$-H), 7.35 (2H, m, C$_{Ar}$-H), 7.60 (12H, d, C$_{Ar}$-H). UV-Vis-NIR (toluene) λ$_{max}$, nm (ε, M$^{-1}$ cm$^{-1}$): 354 (1788), 456 (97). ATIR: 1425.8 (m), 1104.51 (m), 1030.98 (w), 928.16 (w), 744.67 (w), 697.55 (s), 678.99 (m), 559.04 (s), 524.06 (w), 506.92 (m), 483.36 (w), 470.50 (w).

**1U (U$^{IV}$[1,4-(Ph$_3$Si)$_2$C$_8$H$_6$]$_2$(C$_7$H$_8$)):** 74.7% yield as a green powder (60 mg, 38 mol), $^1$H NMR (300 MHz, d$_8$-tol, 350 K) δ (ppm): 5.80 (br s, 12H, C$_{Ar}$-H), 5.44 (br s, 24H, C$_{Ar}$-H), 3.13 (br s, 24H, C$_{Ar}$-H), ring protons not resolvable. UV-Vis-NIR (toluene) λ$_{max}$, nm (ε, M$^{-1}$ cm$^{-1}$): 397 (2243), 592 (1577), 612 (691), 634 (275), 739 (70), 895 (60), 990 (85). ATIR: 1426.52 (m), 1104.51 (m), 1026.69 (w), 930.90 (w), 741.10 (m), 698.98 (s), 679.7 (m), 558.33 (s), 524.77 (w), 507.63 (m), 484.07 (w), 463.37 (w), 427.67 (m).

**1Np (Np$^{IV}$[1,4-(Ph$_3$Si)$_2$C$_8$H$_6$]$_2$(C$_7$H$_8$)):** 77.52% yield as a red powder (10.26 mg, 6.531 μmol), $^1$H NMR (300 MHz, d$_8$-tol) δ (ppm): 6.18 ppm (br s, C$_{Ar}$-H). UV-Vis-NIR (toluene) λ$_{max}$, nm (ε, M$^{-1}$ cm$^{-1}$): 430 (1118), 542 (662), 558 (536), 576 (397), 746 (122), 945 (38).

**1Pu (Pu$^{IV}$[1,4-(Ph$_3$Si)$_2$C$_8$H$_6$]$_2$(C$_7$H$_8$)):** 31.9% yield as a red powder (7.4 mg, 4.7 μmol), $^1$H NMR (300 MHz, d$_8$-tol) δ (ppm): 10.03 ppm (12H, dd, C$_{Ar}$-H), 7.87 ppm (60H, br s, C$_{Ar}$-H). UV-Vis-NIR (toluene) λ$_{max}$, nm (ε, M$^{-1}$ cm$^{-1}$): 411 (1267), 438 (1236), 493 (1210), 564 (907), 941 (126).

### Density functional theory calculations

Geometry optimizations of all **1An** complexes were performed using Kohn–Sham density functional theory (DFT) calculations with ADF version 2023.104[30]. Starting from the experimental crystal structures, both restrained (only hydrogen positions were optimized) and full geometry optimizations were carried out employing PBE0 hybrid functional[31] with 25% exact exchange. Dispersion corrections were not used in the geometry optimizations. All-electron Slater-type orbital (STO) basis sets of triple-ζ doubly polarized (TZ2P) quality have been used for the actinides, and a double-ζ polarized (DZP) quality basis set has been used for other atoms[32]. The choice of the functional is based on previous literature showing the capability to produce an accurate electronic structure and molecular properties in actinide complexes[31,33,34]. Relativistic effects were incorporated using the scalar-relativistic zeroth-order regular approximation (ZORA) Hamiltonian[35]. To incorporate the effects of a solvent, calculations utilized the conductor-like screening model (COSMO) with parameters for toluene, as implemented in ADF[36]. The optimizations were carried out for a $S = 0$, $S = 1$, $S = 3/2$, and $S = 2$ ground spin-state for An = Th, U, Np, Pu, respectively.

Electronic excitation spectra for the complexes were calculated using Gaussian 16 (2016), Revision C.02[37], employing time-dependent density functional theory (TDDFT) response calculations with the Tamm–Dancoff approximation (TDA)[38]. The TDDFT calculations were performed on fully solvent-optimized ground-state geometries of An(COT$^{big}$)$_2$. We note here that all available electronic structure methods that can handle systems as large as the ones studied herein represent a trade-off between accuracy and feasibility. In the context of f-element chemistry, scalar-relativistic time-dependent density functional theory (TDDFT) is a frequently applied tool with known good performance, notwithstanding the inherent multiconfigurational character and spin-orbit coupling that is often present in actinide-containing systems (unless when magnetic properties are of interest)[39,40]. Therefore, despite their approximate nature, DFT and TDDFT remain indispensable tools in the computational chemistry of actinide complexes, particularly for qualitative and semi-quantitative analyses[41,42]. A selection of hybrid density functionals was tested for the present calculations, viz., PBE0, B3LYP, and TPSSh, to assess the functional dependency of the intensity and shift of the UV-Vis peaks. In all these cases, the actinides were treated with SDD valence basis sets accompanied by 60-electron MWB-type effective

core potentials (ECPs)[43–45], and non-metals were treated with def2-SVP basis sets. Solvent effects on the electronic spectra were incorporated via the SMD solvation model[46] with parameters for toluene, corresponding to the solvent used in the spectroscopy experiments. It is noted that, with functional PBE0 and B3LYP, there is an overestimation of intensity in the UV-Vis spectrum along with a shift of *ca.* 0.5 eV and 0.3 eV for **1U** (Fig. S37) relative to the experiment, which is improved with TPSSh functional, leading us to extend our calculations with the TPSSh functional. A total of 300 excitations were calculated for each spectrum, which was more than sufficient to cover the wavelength range explored in the experiments. Selected excitations were analyzed via natural transition orbitals (NTOs)[47].

### Single-crystal X-ray diffraction

Single crystals of the toluene solvate of **1U, 1 Np, 1Pu**, suitable for X-ray diffraction, were obtained by vapor diffusion of hexanes into toluene, while the corresponding benzene solvates were obtained by vapor diffusion of hexanes into a benzene solution. Crystals of **1Th** were all grown from hot solutions. All single crystals of **1An** suitable for X-ray diffraction were mounted on a MiTegen Loop in polyisobutylene oil, while **1Np** and **1Pu** were additionally coated in silicone grease and epoxied to the loop within the inert glovebox and then covered in a MiTegen sheath. The sheath was secured to the base using epoxy. The mounted crystal was then thoroughly decontaminated and surveyed for contamination before being released. All **1An** crystals were collected on a Rigaku XtaLAB Synergy-S diffractometer equipped with a HyPix-6000HE photon counting detector using Cu Kα radiation (λ = 1.54814 Å) (100−240 K). Absorption corrections were done in the CrysAlis Pro (Rigaku Oxford Diffraction) software. The structures were solved using ShelXT[48] and refined using ShelXL[49] in Olex2[50]. Further crystallographic details can be found in the SI.

### UV-Vis spectroscopy

Solution UV-Vis spectroscopy measurements were collected in quartz cuvettes with a 1 cm path length using an Ocean Optics FLAME instrument for **1Np** and **1Pu**, and a Cary 6000i UV-Vis-NIR spectrometer (Agilent Technologies Inc.) controlled with Cary WinUV software for **1Th** and **1U**. Solid-state spectra for **1U** (Fig. S23) were collected on a CRAIC 2030PV PRO UV-VIS-NIR Microspectrophotometer with ~2 nm resolution using single crystals of **1U** on a glass slide with a concave well and sealed with a glass coverslip and epoxy prior to analysis on the instrument.

### Photoluminescence spectroscopy

Photoluminescence spectra were collected in 4 window quartz cuvettes using a Picoquant FluoTime 300 spectrometer for K$_2$COT$^{big}$, **1Th**, and **1U**.

### Infrared spectroscopy

Infrared (FTIR, ATR) spectra of complexes were recorded on a Shimadzu IRSpirit housed in an MBraun glovebox and are reported in wavenumbers (cm$^{-1}$).

### Data availability

Open data, including xyz files for computed structures, and spectroscopic and spectrometric raw data are available at doi: 10.17632/by8ff8pkbx.1. Crystallographic datasets are available from the CCDC deposition numbers 2408395-2408403.

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

## Acknowledgements

The work was supported by the U.S. Department of Energy (DOE), Office of Science, Office of Basic Energy Sciences, Chemical Sciences, Geosciences, and Biosciences Division, Heavy Element Chemistry (HEC) Program at LBNL under contract DE-AC02-05CH11231. C.S.C. was supported under a Department of Energy Nuclear Energy University Programs Graduate Fellowship DE-NE0009073. M.M.P. was supported by the National Science Foundation MPS-Ascend Postdoctoral Research Fellowship under Grant No. 2213284. The isotopes used in this research were from legacy stocks available at Lawrence Berkeley National Laboratory. J.A. acknowledges support of the theoretical component of this study by DOE HEC grant DE-SC0001136.

## Author contributions

Conceptualization: C.S.C., M.M.P., N.J.K., P.L.A. Draft Writing: C.S.C., M.M.P., A.S., N.J.K., J.A., P.L.A. Formal Analysis: C.S.C., M.M.P., A.S., N.J.K. Funding acquisition: P.L.A. Investigation: C.S.C., M.M.P., A.S., N.J.K., M.R.K. Resources: J.J.W., J.A., P.L.A. Supervision: J.A., P.L.A. Reviewing and editing: all authors.

## Competing interests

The authors declare no competing interests.
