## [Peer Review File · Communications Chemistry]

Trends in actinide electronic structure revealed from asymmetric, isostructural transuranic metallocenes

Corresponding Author: Professor Polly Arnold

Version 0:

Reviewer comments:

Reviewer #1

(Remarks to the Author)

This manuscript by Autschbach and Arnold et al. reported synthesis and characterization of a series of actinide metallocenes. The periodic trend of actinides is of interests to the chemistry community. However, the experimental study on actinide complexes, especially for transuranic elements, is restricted by the radioactive nature of these elements as well as regulations. Therefore, it is always welcomed to see experimental contributions. Overall, this paper will be timely and welcomed to the community. In addition, the manuscript is easy-to-read and digest. Following are some suggestions to improve the paper before it may be accepted for publication.

1. In addition to percentage yield, the absolute yield (in g or mg) should be provided.
2. For DFT calculations, it would be beneficial if the authors can provide the Kohn-Sham orbitals in the main text to show bonding interactions between actinide ions and COTbig ligands. Some additional discussion about the contributions from metal-based orbitals is also useful.
3. What is the implication of "the sum of the remaining charges" in Table 2? Since COT and COTbig have very different atoms, this seems meaningless.
4. "The multiple peaks observed in K2COTbig and the shoulders in 1Th are assigned as vibronic coupling to the IR modes observed at 1427 and 1426 cm⁻¹ respectively." Please explain or provide reference for reasoning.
5. Please define hexane as either n-hexane or hexanes in the main text.

Reviewer #2

(Remarks to the Author)

The manuscript by Conour et al. describes the synthesis of a series of [An(COTbig)₂] complexes for An = U, Th, Np and Pu and their experimental and computational characterisation. All compounds were characterised by single crystal XRD, ¹H NMR and UV/Vis spectroscopy. The Th and U complexes were additionally characterised by photoluminescence and IR spectroscopy. Additionally, DFT and TDDFT calculations were performed to support the analysis of UV/Vis spectra. The authors demonstrate the effect of the substitution at the COT ligands on the structure, electronic properties and thus the spectroscopic observations in comparison to previously reported actinocenes.

The manuscript is generally written well and the results are presented clearly. The authors analysed the data thoroughly and especially the combination of spectroscopic and computational analyses yields in-depth insights into changes in the bonding situation across the studied series. The presented study marks a valuable contribution to the understanding of trends in the electronic structure amongst the actinoids. I would therefore like to recommend the acceptance of the manuscript for publication in Communications Chemistry after a few issues have been addressed.

Main manuscript:

- I don't agree with the second sentence of the introduction: apart from studies on actinyl ions, a large portion of the understanding of bonding (in terms of covalency vs. ionicity) in actinoid complexes has been gained from non-aqueous studies and this is also the main point of the cited review articles. Please clarify your statement.
- The authors note the presence of rotational isomers in solution for 1Th and 1U, while only one of the isomers is observed in the solid state, which is attributed to dispersion. Was this effect evaluated in the DFT study? I did not see any remarks on inclusion of dispersion in the DFT methods. Were the anti-isomers calculated as well and if so, does the UV/Vis spectrum change?
- Figure 3: please expand the range for the U spectra to include the signal at 3.5 eV.
- Methodology section: it would be helpful to the reader to refer to the SI for the extended experimental details, or, if space permits, include the full synthesis details here.

Supporting information:

- In the pdf file some figures got lost in comparison to the .docx file (S 27, S28, S30, S32, S33, S34).

- Fig. S31: please include the NTOs for the transition at 2.8 eV.
- Fig. S33: the assignment which panel (a, b, c, d) is for which metal is missing.
- Please include the xyz data for the optimised structures either in the SI or as .xyz files or via a repository.

Reviewer #3

(Remarks to the Author)

"Trends in actinide electronic structure revealed from asymmetric, isostructural transuranic metallocenes"

The manuscript reports the synthesis and characterization of a novel series of actinide metallocene complexes, $An(COTbig)_2$ ($An = Th, U, Np, Pu$), featuring bulky 1,4-bis(triphenylsilyl)-substituted cyclooctatetraenyl (COTbig) ligands. These complexes exhibit a bent metallocene structure, distinct from the known coplanar $An(COT)_2$ analogs, enabling the exploration of electronic structure trends across the actinide series. Combining synthetic chemistry, structural analysis (including SCXRD), spectroscopic studies (UV-vis, photoluminescence, IR), and computational (DFT and TDDFT) modeling, the authors aim to elucidate the roles of 5f and 6d orbitals and highlight changes in covalency and bonding trends across Th-Pu. Notably, they attribute increased low-energy f-f transition intensities in $Pu(COTbig)_2$ to enhanced covalency and ligand-metal orbital mixing enabled by the bent geometry and unsymmetric ligand substitution.

Major Comments

The synthesis of a complete isostructural series of $An(COTbig)_2$ complexes including Th, U, Np, and Pu is an important advancement, given the difficulty in achieving homoleptic, isostructural sets of transuranic organometallics. The introduction of a bent geometry, breaking the inversion symmetry present in $An(COT)_2$, provides a unique platform to probe actinide-ligand orbital interactions, particularly f-orbital contributions. The combination of experimental and computational analyses makes a valuable contribution to understanding electronic structure trends across the actinide series. This represents a notable and impactful advance suitable for *Nature Communication*, particularly due to its implications for f-element bonding and actinide reactivity.

However, some conclusions appear speculative or insufficiently supported. For example, while the claim that the removal of inversion symmetry enhances f-f transition intensities is plausible, it lacks direct evidence (e.g., quantitative transition intensity analysis or complementary spectroscopic techniques). Similarly, the assertion that the increased molar absorptivity of $Pu(COTbig)_2$ arises from enhanced 5f- π mixing would benefit from more direct comparison to experimental data or clearer computational evidence. Explicit discussion of the limitations of the models (e.g., TDDFT's known challenges for actinides) would strengthen the credibility of the conclusions.

Experimental and Computational Methods

The computational approach employs PBE0 and TPSSH functionals with ZORA relativistic corrections and appropriate basis sets/ECPs. However, some computational assumptions should be explicitly justified, such as why spin-orbit coupling (SOC) is not included in DFT or TDDFT despite its known importance for actinides. In particular, it seems necessary to include SOC when assessing the accuracy of the different functionals B3LYP, PBE0, TPSSH. Moreover, the redshift and blueshift corrections applied to match computed and experimental spectra suggest limitations of the model that are not fully acknowledged.

Overall, the manuscript is well-organized and clearly written, but some arguments lack full support:

- The connection between bent geometry and enhanced f-f transitions (p.8, lines 215–225) is logical but lacks direct spectroscopic evidence (e.g., a clear comparison of transition intensities with inversion symmetry). A quantitative analysis of transition intensities for both coplanar and bent complexes would strengthen this claim.
- The assignment of photoluminescence peaks (1Th vs. $K_2(COTbig)$) is plausible, but the explanation for the decreased fluorescence upon binding to Th is speculative and would benefit from fluorescence lifetime measurements or additional controls.
- The electronic structure trends are presented in detail (Table 2), but the discussion of why ligand-metal donation is strongest for U/Np and weakest for Th (p.9, lines 244–246) is somewhat superficial, attributing trends to orbital energy and overlap without rigorous analysis. Including orbital interaction diagrams or energy level schemes would help clarify these trends.

Recommendation

I recommend moderate revision before acceptance. The work is novel and significant, with the potential to make a high-impact contribution to actinide chemistry. However, to meet the rigor and conclusiveness expected for *Nature Communications*, the authors should revise the manuscript according to the comments pointed out above.

Version 1:

Reviewer comments:

Reviewer #1

(Remarks to the Author)

I am fine with the response from the authors. The paper is now acceptable for publication.

Reviewer #2

(Remarks to the Author)

Thank you for the corrections and replies, I am happy for the revised manuscript to be accepted as it is.

Reviewer #3

(Remarks to the Author)

I would like to thank the authors for their thoughtful and thorough revision. I appreciate the care they have taken in addressing the comments and suggestions. The changes made significantly improve the clarity and quality of the manuscript, and I am satisfied with how you have incorporated the feedback.

Reply to Reviewers

Reviewer 1

R1.1 In addition to percentage yield, the absolute yield (in g or mg) should be provided.

Response: These data (percents and absolute) were included in the SI, but we have updated the main manuscript to include these.

R1.2 For DFT calculations, it would be beneficial if the authors can provide the Kohn-Sham orbitals in the main text to show bonding interactions between actinide ions and COTbig ligands. Some additional discussion about the contributions from metal-based orbitals is also useful.

Response: We now show selected Kohn-Sham orbitals (HOMO) evidencing the “delta”-bonding interaction between the ligand and the actinides. A brief description of these orbitals evidencing the donation mode was added to the manuscript text.

R1.3 What is the implication of “the sum of the remaining charges” in Table 2? Since COT and COTbig have very different atoms, this seems meaningless.

Response: This phrasing choice was leftover from a previous manuscript edit. Apologies, it has been removed.

R1.4 “The multiple peaks observed in K2COTbig and the shoulders in 1Th are assigned as vibronic coupling to the IR modes observed at 1427 and 1426 cm⁻¹ respectively.” Please explain or provide reference for reasoning.

Response: The peaks in one absorption envelope of the vibrational spectra are split due to vibronic coupling (thermal population of a set of vibrational modes that are close in energy). We have reworded this for clarity.

R1.5 Please define hexane as either n-hexane or hexanes in the main text.

Response: We thank the reviewer for noting this inconsistency. It has now been fixed in the main text and supporting information.

Reviewer 2

Main manuscript:

R2.1 I don't agree with the second sentence of the introduction: apart from studies on actinyl ions, a large portion of the understanding of bonding (in terms of covalency vs. ionicity) in actinoid complexes has been gained from non-aqueous studies and this is also the main point of the cited review articles. Please clarify your statement.

Response: Thank you for challenging us. We have now clarified the statement in the introduction, specifying the predominance of aqueous studies in *historical* actinide research while highlighting the more recent efforts towards non-aqueous chemistry.

R2.2 The authors note the presence of rotational isomers in solution for 1Th and 1U, while only one of the isomers is observed in the solid state, which is attributed to dispersion. Was this effect evaluated in the DFT study? I did not see any remarks on inclusion of dispersion in the DFT methods. Were the anti-isomers calculated as well and if so, does the UV/Vis spectrum change?

Response: The precise reasons for the preference of the isomers in the solid state are not known. “Dispersion” in the context of crystal structures should therefore have referred to solid state packing effects, which is now clarified. Empirical dispersion corrections were not included during the geometry optimization of the complexes. This is now explicitly stated. Previous studies involving COT-based ligands by our team have shown that the incorporation of empirical dispersion corrections can lead to poor agreement with experimental structural parameters. While compounds **1Th** and **1U** exhibit some degree of rotational isomerism, this aspect is beyond the scope of the present investigation. Thanks to this interesting question, we have returned to further experiment efforts to determine the impact of the presence of an anti-isomer in solution on the UV-vis spectra. To do this, we collected solid-state UV-vis data on a single crystal sample of the *gauche* isomer of **1U**. This spectrum overlaps very well with the solution-state spectrum which contains the “mixture” of isomers that will be accessible at room temperature on the IR timescale. Therefore we conclude that the presence of an *anti* isomer has minimal impact on the electronic spectra of these complexes.

R2.3 Figure 3: please expand the range for the U spectra to include the signal at 3.5 eV.

Response: We have widened the spectral range in the manuscript.

R2.4 Methodology section: it would be helpful to the reader to refer to the SI for the extended experimental details, or, if space permits, include the full synthesis details here.

Response: We have added text to refer the reader to the SI for these.

Supporting information:

R2.5 In the pdf file some figures got lost in comparison to the .docx file (S 27, S28, S30, S32, S33, S34).

Response: We are sorry the reviewer had a problem with the PDF. We have converted the edited documents to pdf and viewed it on different computers with no issues. We will take this care with the uploaded revisions.

R2.6 Fig. S31: please include the NTOs for the transition at 2.8 eV.

Response: We have added the NTOs for the transition at 2.8 eV in the revised supporting information.

R2.7 Fig. S33: the assignment which panel (a, b, c, d) is for which metal is missing.

Response: The panel labels in Fig. S33 (now Fig S34) are now explained in the caption of the figure.

R2.8 Please include the xyz data for the optimised structures either in the SI or as .xyz files or via a repository.

Response: Happy to. These are in the open data archive, which is now cited in the manuscript. Arnold, Polly (2025), “AnCOTbig”, Mendeley Data, V1, doi: 10.17632/by8ff8pkbx.1

Reviewer 3

Major Comments

R3.1 However, some conclusions appear speculative or insufficiently supported. For example,

while the claim that the removal of inversion symmetry enhances f-f transition intensities is plausible, it lacks direct evidence (e.g., quantitative transition intensity analysis or complementary spectroscopic techniques).

Response: We thank reviewer three for pointing out this example. While we would have traditionally directly compared the two, this is challenging as we cannot observe the corresponding f-f transitions in the original $\text{An}(\text{COT})_2$ series. We suggest that this lends credence to the fact that the intensities of this transition are enhanced *via* the COT^{big} system.

R3.2 Similarly, the assertion that the increased molar absorptivity of $\text{Pu}(\text{COT}^{\text{big}})_2$ arises from enhanced $5f-\pi$ mixing would benefit from more direct comparison to experimental data or clearer computational evidence.

Response: We believe it is reasonable to expect that enhanced $5f-\pi$ mixing will to increased molar absorptivity, as is seen in d-block chemistry where d- π mixing has the analogous effect. Unfortunately, the computational tools to definitively prove this assertion are still in development (please see our response to R3.7). We have replaced the language of assertion with that of suggestion in the MS.

R3.3 Explicit discussion of the limitations of the models (e.g., TDDFT's known challenges for actinides) would strengthen the credibility of the conclusions.

Response: We have now added a brief discussion of the necessarily approximate nature of DFT/TDDFT for actinides in the revised manuscript (Density Functional Theory Calculations section), along with additional relevant references.

Experimental and Computational Methods

R3.4 The computational approach employs PBE0 and TPSSh functionals with ZORA relativistic corrections and appropriate basis sets/ECPs. However, some computational assumptions should be explicitly justified, such as why spin-orbit coupling (SOC) is not included in DFT or TDDFT despite its known importance for actinides. In particular, it seems necessary to include SOC when assessing the accuracy of the different functionals B3LYP, PBE0, TPPSh. Moreover, the redshift and blueshift corrections applied to match computed and experimental spectra suggest limitations of the model that are not fully acknowledged.

Response: All calculations on metal complexes as big as the systems studied by us require approximate methods. This is now explicitly stated in the manuscript (see our reply to the previous comment). The functional comparisons that were performed in our study represent the assessment mentioned by the reviewer, showing that TPSSh performs best, in keeping with other DFT/TDDFT f-element studies (see, for example, the extensive body of work using scalar relativistic DFT/TSSDT by W. Evans, F. Furche, et al, other studies by our team, and many other publications). The spectral shifts that were applied are rather standard in this field of research and therefore not particularly noteworthy.

R3.5 The connection between bent geometry and enhanced f-f transitions (p.8, lines 215–225) is logical but lacks direct spectroscopic evidence (e.g., a clear comparison of transition intensities with inversion symmetry). A quantitative analysis of transition intensities for both coplanar and bent complexes would strengthen this claim.

Response: We thank the reviewer for this comment, but unfortunately cannot perform a quantitative analysis of the f-f transitions between the coplanar and bent complexes, as noted in our response to R3.1. However, we have changed our wording in the manuscript to clarify that

the bent nature of the system is a *possible* reason for the enhanced intensity, but has not been experimentally determined to be the sole cause.

R3.6 The assignment of photoluminescence peaks (1Th vs. K₂COT^{big}) is plausible, but the explanation for the decreased fluorescence upon binding to Th is speculative and would benefit from fluorescence lifetime measurements or additional controls.

Response: We appreciate the reviewers comment on this point, unfortunately we do not currently possess the capability to perform lifetime studies at excitation wavelengths appropriate for **1Th**. As a result, we have struck the comment from the manuscript in the absence of further experimental evidence.

R3.7 The electronic structure trends are presented in detail (Table 2), but the discussion of why ligand–metal donation is strongest for U/Np and weakest for Th (p.9, lines 244–246) is somewhat superficial, attributing trends to orbital energy and overlap without rigorous analysis. Including orbital interaction diagrams or energy level schemes would help clarify these trends.

Response: We appreciate the comments. The exact “Why?” posed by the reviewer has mystified the community for decades. The general lowering of the 5f-shell energy along the actinide series, used by us to rationalize some of the observed trends, is an established fact and does not require further proof. However, the necessary tools to evaluate the precise roles of overlap and AO energy matching explicitly are only now under development (in the group of one of us). We plan to evaluate the present COT^{big} complexes as part of a broader study of f-element covalency eventually. At present these methods are not published (an initial proof of concept for high-symmetry systems is under review) and requires more testing before they can be applied to complex systems. Therefore, no changes were made to the manuscript in response to this comment by the Reviewer.